# Examining Insula–Default Mode Network Functional Connectivity and Its Relationship with Heart Rate Variability

**DOI:** 10.3390/brainsci15010037

**Published:** 2025-01-01

**Authors:** Marlene Nogueira, Juliana da Silva Magalhães, Adriana Sampaio, Sónia Sousa, Joana F. Coutinho

**Affiliations:** Psychological Neuroscience Laboratory, Psychology Research Centre (CIPsi), School of Psychology, University of Minho, 4710-057 Braga, Portugal; marlene.nogueira@psi.uminho.pt (M.N.); julianamsm@gmail.com (J.d.S.M.); adriana.sampaio@psi.uminho.pt (A.S.); d7879@psi.uminho.pt (S.S.)

**Keywords:** insula, default mode network, functional connectivity, heart rate variability

## Abstract

Background: The Default Mode Network (DMN) is involved in self-referential and emotional processes, while the insula integrates emotional and interoceptive signals. Functional connectivity between the insula and the DMN is proposed to enhance these processes by linking internal bodily states with self-referential thoughts and emotional regulation. Heart Rate Variability (HRV), a measure of parasympathetic regulation of cardiac activity, has been associated with the capacity to regulate autonomic arousal. This study explored the relationship between insula–DMN functional connectivity and HRV. We hypothesized that (1) insula’s functional activity and volume would be related to HRV; (2) insula activation would be functionally connected with the DMN; and (3) stronger insula–DMN connectivity would correlate with higher HRV. Methods: Forty-three healthy adults underwent a structural and functional MRI acquisition to assess insula–DMN connectivity during resting state. HRV was measured also at rest using the BIOPAC system. Results: A significant positive correlation was found between insula–DMN connectivity, but no correlation was observed between insula–DMN connectivity and HRV. We also found a positive significant association between left insula volume and HRV. Conclusions: These findings suggest that while the AI and DMN are functionally interconnected, this connectivity may not be directly related to HRV. The results highlight the complexity of the relationship between brain connectivity and autonomic function, suggesting that other factors may influence HRV.

## 1. Introduction

The insular cortex, often referred to as the insula, is a pivotal yet discreet segment of the cerebral cortex. Situated within the Sylvian fissure, the insula has a triangular structure composed of both short and long gyri and is covered by the opercula of the frontal, parietal, and temporal lobes.

Central to interoceptive functions, the insula regulates our perception of internal sensations such as thirst, hunger, and pain [1,2]. It also plays roles in motor functions [3], emotional processing within the limbic system [4], and cognitive tasks including attention, decision-making, and time perception [5]. Additionally, it aids social cognition and autonomic functions, calibrating vital metrics like heart rate and blood pressure [6]. This integrative role links bodily signals with emotions, self-awareness, cognitive control, and decision-making.

The insula’s intricate functional connectivity with the amygdala, Anterior Cingulate Cortex (ACC), and prefrontal cortex highlights its key role in merging internal bodily information with cognitive and emotional processes [5]. Through these networks, the insula enables top-down control over emotional responses, modulating how individuals react to emotional and interoceptive signals [7], through distinct autonomic functions. The insula, particularly its anterior segment, plays a central role in influencing heart rate and heart rate variability (HRV), mediated through the central autonomic network, encompassing the amygdala, hypothalamus, and brainstem, which governs autonomic functions and stress responses [8] and by integrating cardiac signals with emotional and cognitive information [2,9]. Specifically, in emotional contexts, the insula’s involvement is especially salient, modulating emotional arousal and autonomic responses to varying stimuli and stressors, through its modulation of heart rate and HRV [9]. Through this integration, the insula helps maintain autonomic balance, supporting homeostasis and adaptive responses to emotional and physiological demands [2,10].

The anterior insula is intricately tied to interoception, emotional processing, and physiological regulation, notably concerning heart rate [11,12].

Research by Jones et al. (2015) investigated how individuals could modulate their heart rate using biofeedback techniques, specifically examining the neural activity involved in this process [13]. The study utilized fMRI to observe brain activity while participants actively attempted to raise or lower their heart rate through biofeedback. The results revealed lateralized activity in the insula, with the right anterior insula predominantly active during heart rate elevation and the left anterior insula more engaged during heart rate reduction. In addition to the anterior insula, other brain regions such as the dlPFC and the ACC were also involved, demonstrating a network of regions that interact to modulate heart rate. This suggests that heart rate modulation is supported by a broader neural consortium, with the insula playing a central role in integrating emotional and physiological signals.

The insula’s involvement in emotional processing intersects with its role in heart rate regulation, suggesting an intertwined relationship. HRV, which refers to variations in the time intervals between successive heartbeats, is a marker of emotional health [14,15]. Higher HRV indicates robust emotional regulation capabilities and resilience to stressors [15].

Thayer et al. (2012) conducted a meta-analysis highlighting the roles of the amygdala, vmPFC, and ACC in HRV and emotional regulation [16]. Although the insula was not directly cited, its robust connectivity with these regions suggests its complementary role in HRV and emotional regulation. On the other hand, previous work has shown the association between HRV and insula structural measures (Matusik et al., 2023) [17].

The anterior insula (AI), with its role in processing salient emotional stimuli and interoception, plays a complementary role with the default mode network (DMN) to integrate these signals with self-referential thought processes. This interplay ensures that emotional experiences are contextualized within a personal narrative, facilitating effective emotional regulation [5,18].

The DMN, comprising brain regions such as the medial prefrontal cortex (mPFC), posterior cingulate cortex (PCC), precuneus, and lateral parietal cortex [19] is predominantly active during rest and self-referential processes, such as mind-wandering, introspection, and recalling personal memories [20]. It integrates past experiences and personal relevance into the current emotional context, which is essential for adaptive emotional regulation [21].

The interaction between the DMN and the insula is pivotal for aligning interoceptive signals with emotional and cognitive processes. While the insula processes interoceptive signals and salient emotional stimuli, the DMN contextualizes these signals within the framework of self-referential thoughts and memories. This integration allows individuals to reflect on their internal states and adjust their emotional responses accordingly [5]. For example, the mPFC, a key node in the DMN, has been shown to interact with the insula to balance internal emotional states with external situational demands [18], a critical component of effective emotional regulation.

Additionally, the DMN’s involvement in constructing and maintaining a coherent self-narrative provides a stable background against which emotional experiences are evaluated and regulated [22]. Therefore, the insula–DMN connectivity underscores a fundamental mechanism by which the brain integrates internal bodily states with cognitive evaluations of these states to maintain emotional balance.

Although prior research has demonstrated links between the insula or the DMN and cardiac function, few studies have explored their combined influence on HRV. For example, in the study by Chang et al. (2013) [23], fluctuations in HRV were associated with changes in the functional connectivity of the insula and other brain regions, suggesting that autonomic regulation (as indicated by HRV) is linked to brain network activity, particularly during resting states. The insula, known for its role in autonomic regulation and emotional processing, showed significant connectivity with regions involved in self-regulation and emotional control (Chang et al., 2013) [23]. Yoshikawa et al. (2020) [24] investigated how heart rate affects the functional connectivity of the DMN and found that higher HRV was associated with stronger connectivity within the DMN. This suggests that individuals with better autonomic regulation (as indicated by higher HRV) may have more stable and efficient functioning of the DMN (Yoshikawa et al., 2020) [24].

Additionally, the functional connectivity between the DMN and the insular cortex has been explored in the context of various disorders (e.g., Hsiao et al., 2017; Tsurumi et al., 2020) [25,26], highlighting their role in emotional and cognitive processes. However, to the best of our knowledge, the combined relationship between insula–DMN connectivity and HRV has not been specifically addressed. Moreover, structural measures of the insula, such as gray matter volume, have been correlated with HRV (Matusik et al., 2023; Wei et al., 2018) [17,27]. These findings suggest that the insula’s structural and functional characteristics are integral to autonomic control, but the role of its connectivity with the DMN in this context remains underexplored.

### Research Problem and Hypothesis

The insula’s multifaceted functions span interoception, emotional regulation, and autonomic control, with its structural integrity reflected in gray matter volume (GMV). Studies suggest that greater GMV in the insula correlates with enhanced neural resources for autonomic regulation, including heart rate variability (HRV (Matusik et al., 2023) [17], which is a biomarker of autonomic nervous system activity, that reflects autonomic stability and emotional health [11,12,15].

The DMN’s role in maintaining a coherent self-narrative and contextualizing interoceptive signals facilitates autonomic balance by modulating parasympathetic responses. This modulation supports the maintenance of vagal tone, a critical component of autonomic regulation. The collaboration between the DMN and AI is particularly relevant in managing bodily states and achieving this balance [16]. However, the direct relationship between the functional connectivity of the insula and DMN with HRV remains less explored.

Despite substantial evidence on the individual roles of the insula and DMN in emotional and autonomic regulation, their joint contribution to HRV remains underexplored. Functional connectivity between these regions may represent a crucial mechanism by which the brain integrates interoceptive and self-referential processes to regulate autonomic balance.

Therefore, this study aims to investigate the functional connectivity between the insula and the DMN, and its relationship with HRV. We hypothesize that: (1) the insula’s activity and volume would be related to HRV; (2) the insula activation would be functionally connected with the DMN; and (3) stronger insula–DMN connectivity would correlate with higher HRV. By exploring these hypotheses, this study seeks to deepen the understanding of how these brain regions coordinate autonomic and emotional processes, potentially revealing the neural mechanisms underlying adaptive emotional regulation.

## 2. Materials and Methods

### 2.1. Participants

Forty-three healthy adults (23 males and 20 females), Caucasian and right-handed with ages ranging from 23 to 39 years (M = 31.14, SD = 4.54) participated in this study. Participants were screened based on the following exclusion criteria: (1) any diagnosis of dementia, neuropsychiatric, or neurodegenerative disorders; (2) history of alcohol or drug dependence or abuse within the past year; (3) inability to attend MRI sessions due to conditions such as metallic implants or pregnancy; (4) age below 20 or above 50 years. The study complied with the principles expressed in the Declaration of Helsinki (with the amendment of Tokyo 1975, Venice 1983, Hong Kong 1989, Somerset West 1996, Edinburgh 2000) and was approved by the local University Institutional Review Board (Code: CIPSI/CE/2013/02). All participants were fully informed about the study’s purpose, procedures, and any potential risks and provided written informed consent. Confidentiality and anonymity were maintained throughout the study. Data were stored securely, and access was restricted to authorized personnel only.

### 2.2. Structural and Functional MRI Acquisition

Structural (T1) and functional images (T2) were acquired with a clinically approved 3T MRI scanner (Siemens Magnetom Tim Trio, Erlangen, German). Before the MRI acquisition participants were screened for objects that could interact with the scanner magnetic field (e.g., earrings, keys, etc.) and affect their safety. Participants were then familiarized with the scanner and instructed on the procedure. Participants were instructed to keep their eyes closed, stay awake, and remain as still as possible to minimize motion-related artifacts.

Structural T1 scans were acquired with the following parameters: 192 sagittal slices, repetition time (TR) = 2000 ms; echo time (TE) = 2.33 s, flip angle = 7°, slice thickness = 0.8 mm, slice gap = 0 mm, pixel size = 0.8 × 0.8 mm^2^, field of view (FoV) = 256 mm.

The resting-state fMRI acquisition had a duration of seven minutes. The images were acquired using a blood oxygen level dependent (BOLD) sensitive echo-planar imaging (EPI) using the following parameters: repetition time (TR) of 2000 ms, echo time (TE) of 29 ms, flip angle of 90°, field of view (FoV) of 222 × 222 mm, voxel size of 3 × 3 mm with 39 slices and 210 volumes.

### 2.3. Image Preprocessing

#### 2.3.1. Structural Images

After the acquisition, the structural images were reconstructed and analyzed using FreeSurfer (http://surfer.nmr.mgh.harvard.edu/ (accessed on 1 April 2017), a widely used software, and a well-developed application. (Hodneland et al., 2012) [28]. A manual verification of the reconstructed images consisted of verification of the coregistration in the Tailarach template, which is based on the distance between the anterior (AC) and posterior commissure (PC); verification of skull removal from the images, to correct for mistakes of brain tissue and verification of the limits of the cortical and subcortical areas [29]. After this verification, we obtained the processed images, allowing for the application of a consistent protocol of GM identification in the ROI of interest: left and right insula. The intracranial total volume (ITV) was calculated using the statistics from Free Surfer.

#### 2.3.2. Functional Images

Image preprocessing was performed using FMRIB Software Library (FSL v5.09). The preprocessing steps included the removal of the first five volumes to stabilize the signal, slice-timing and motion correction using rigid body alignment. Motion scrubbing was also performed, removing volumes where frame displacement and DVARS exceeded the set thresholds, to minimize motion-related artifacts. The fMRI data were normalized to the MNI standard space and further processed through regression of motion parameters and mean white matter and cerebrospinal fluid signals. Band-pass temporal filtering (0.01–0.08 Hz) and spatial smoothing with an 8 mm full-width at half-maximum Gaussian kernel were applied to enhance signal clarity. All data acquisitions underwent visual inspection to ensure they were free from undue head motion or brain lesions.

The data quality was verified before analysis, and the imaging quality was confirmed as appropriate for subsequent analyses.

### 2.4. Cardiac Measures

Heart rate variability (HRV) was measured using the BIOPAC system. Participants had abstained from alcohol, nicotine, and caffeine for at least four hours prior to the data collection. Cardiac data were registered at resting-state to provide reference values of cardiac activity for each participant, in order to establish a baseline HRV, that could serve as a biomarker for autonomic nervous system activity and emotional regulation capacity. The data acquisition involved recording the electrocardiogram (ECG) signal with high temporal resolution during the baseline phase at a 1000 Hz sampling rate, during approximately six minutes.

A 3-electrode Lead-II configuration was used to collect the raw ECG, with electrodes placed on the right and left mid-clavicle and a third on the upper left shoulder. Skin preparation involved cleaning with alcohol and drying to minimize impedance, ensuring high-quality electrophysiological signals. Nonpolarizable disposable Ag–AgCl electrodes (1 cm diameter contact area, EL-503, BIOPAC Systems, Santa Barbara, CA, USA) were employed. The RSPEC-R module of the BIOPAC BioNomadix wireless system [30] interfaced with the MP150-BIOPAC data acquisition system to capture cardiac activity. The receiver was linked to a host computer running AcqKnowledge software (version 4.4). The ECG data underwent standard filtering with a 1 Hz IIR high-pass filter and a 35 Hz low-pass filter. Visual inspection was conducted to identify ectopic beats and artifacts, with manual correction applied when the automated QRS detector misidentified R-spikes [31]. To measure HRV we used a frequency domain approach that allows for the partition of the total variability into frequency components, thus allowing the identification of specific components of heart rate variability. The band-limited variance for the HF-HRV index was extracted through the automated analysis tool provided by the manufacturer which uses as a spectral analysis approach the Fast Fourier Transformation analysis (FFT) method, according to the international cardiology standards [32,33]. HF-HRV (high-frequency- HRV) was calculated offline using Acknowledge software (Version 4.4; Biopac Systems). HF-HRV was recorded as an average value per individual.

### 2.5. Data Analysis

Normality tests were conducted to ensure the suitability of parametric methods for analyzing the relationship between brain measures and HRV. The Shapiro–Wilk test results indicated that the distributions of the variables did not significantly deviate from normality (*p* > 0.05), validating the assumption of normality. 

### 2.6. Insula–DMN Functional Connectivity

To understand the dynamic interplay between DMN and insula, the functional connectivity between these brain regions was assessed. Following the verification of imaging quality, fMRI scans were processed to extract the temporal series data for the insula and the DMN using FMRIB Software Library (FSL) fslmeants tool.(FSL v5.09; http://fsl.fmrib.ox.ac.uk/fsl/, accessed on 24 November 2024). This step was essential to assess the functional connectivity between these regions. Preprocessed fMRI data, stored as a 4D volume file was used as the input. Region-specific masks for the insula and the DMN were applied to isolate the signals of interest (Figure 1). The output generated was a time series for each participant, capturing the temporal fluctuations of BOLD signals within the respective regions. The fslmeants commands used in this study are available upon request.

### 2.7. HRV Relationship with Insula’s Morphometry and Resting-State Activity

To assess the relationship between insula’s right, left and total volumes and HRV, Pearson correlation analyses were conducted. Pearson’s correlation coefficient measures the strength and direction of the relationship between two continuous variables. HRV were treated as the independent variable, while insula volumes (morphometry analysis) and time series (resting-state activity) were treated as dependent variables.

To further assess the relationship between HRV and the functional activity of the insula, Pearson correlation analyses were conducted using the mean time series values extracted from the insula region of interest (ROI) during resting-state activity. HRV measure was treated as the independent variable, while the mean time series values from the insula were treated as the dependent variable.

The correlations were interpreted using Cohen’s guidelines: coefficients of r > 0.5 indicate a strong relationship, 0.3 ≤ r ≤ 0.5 indicate a moderate relationship, and r < 0.3 indicate a weak relationship. Statistical significance was assessed at a *p* < 0.05 threshold. All analyses were performed using Jamovi software (version 2.5, https://www.jamovi.org (accessed on 1 September 2024)) [34,35].

## 3. Results

### 3.1. Insula and DMN Functional Connectivity

A significant correlation was observed between insula and DMN, functional connectivity (r = 0.493, *p* = 0.003, n = 35), indicating a strong positive relationship between the two variables.

The temporal series graph (Figure 2), presented below, illustrates the fluctuations in functional connectivity between the insula and the DMN throughout the scan acquisition time. Notably, this graph reveals synchronous patterns and variations in connectivity.

### 3.2. Insula Morphometry and HRV

The Pearson’s correlation analysis revealed a significant positive correlation between the HRV, and left insula volumes (r = 0.365, *p* = 0.016, n = 43) (see Table 1).

### 3.3. Insula Resting-State Activity and HRV

The Pearson’s correlation analysis revealed no significant correlation between the HRV and insula resting-state activity (r = 0.244, *p* = 0.158, n = 35) (see Figure 3).

### 3.4. Insula–DMN Connectivity and HRV

A Fisher transformation was applied to the insula–DMN correlation values to normalize the insula–DMN connectivity variable, allowing it to be suitable for correlational analysis with HRV.

The results indicated that there was no significant correlation between the Fisher-transformed insula–DMN connectivity values and the HRV values (r = 0.004; see Table 1), suggesting that the insula–DMN connectivity was not directly associated with HRV in this sample. Figure 4 illustrates the scatter plot of the Fisher-transformed insula–DMN connectivity values against HRV, further demonstrating the absence of a significant trend (with R^2^ = 0.002).

## 4. Discussion

The main goal of this work was to examine the interplay between brain regions and networks involved in emotional processing and regulation like the insula and the DMN and cardiac measures associated with autonomic regulation, specifically with parasympathetic (or vagal) influences over the heart such as the HRV.

Regarding our first hypothesis we found an association between the volume of left insula and HRV. The lateralization of these results is in line with the previously mentioned study by Jones et al. (2015) that used fMRI to observe brain activity while participants actively attempted to raise or lower their heart rate through biofeedback [13]. They found lateralized activity in the insula, with the right anterior insula predominantly active during heart rate elevation and the left anterior insula more engaged during heart rate reduction. In the same direction are the results reported by Lane et al. (2009) who examined the neural correlates of HF-HRV during emotion in real time and found that HF-HRV was correlated with emotionally induced brain activation in left mid-insula as well as in the caudate nucleus and periaqueductal gray [36]. The authors argued that the association between HF-HRV and activity in the left insula is consistent with its well-known role in emotion and autonomic regulation. Indeed, the vagal (high frequency HF) component of heart rate variability (HRV) is considered to reflect vagal antagonism of sympathetic influences, thus the observed associations association between left insula and HRV at the volumetric level may suggest the possibility of a lateralization in this brain structure. These results support the view that the medial visceromotor network is a final common pathway by which emotional and cognitive functions recruit autonomic support.

This study revealed a significant positive correlation between the functional connectivity of the insula and the DMN, indicating that increased functional connectivity within the DMN is associated with a corresponding increase in connectivity within the insula.

Despite the robust association, no significant relationship was observed between insula–DMN connectivity and HRV. This result contradicts the initial hypothesis, which posited that stronger insula–DMN connectivity would predict higher HRV, a marker often linked to emotional regulation capacity. While the insula and the DMN exhibit strong functional connectivity, this neural interplay does not appear to directly influence autonomic regulation, as measured by HRV, in this sample.

Insula plays a crucial role in integrating autonomic and emotional processes, acting as a hub that connects bodily states with cognitive-emotional awareness [9]. Studies have shown that the insula’s connectivity with the DMN is integral in self-referential thinking and internal bodily awareness, both of which are fundamental for emotional regulation [5]. The strong functional connectivity between the insula and the DMN observed in the present study aligns with previous research suggesting that the insula facilitates the transition between self-referential mental states, mediated by the DMN, and externally focused tasks through its involvement in the salience network [18]. This relationship highlights the anterior insula’s ability to modulate the DMN based on emotional and autonomic demands, which is critical for adaptive emotional responses. However, our findings also suggest that this connectivity may not translate directly to autonomic regulation as captured by HRV, hinting that the insula–MN axis may influence higher-order cognitive-emotional processes without directly modulating autonomic responses, as reflected by physiological markers such as HRV.

The absence of correlation between insula-DMN connectivity and HRV suggests that other mediating factors may play a critical role in autonomic function. It is possible that HRV, as a physiological marker, is influenced by mechanisms beyond the scope of the connectivity patterns between these two regions. The anterior insula’s role in autonomic control, as previously established in literature [2,9] may operate through pathways or networks not fully captured in the scope of this study, which primarily focused on the DMN. The DMN’s involvement in self-referential thought processes and contextualization of emotional states [5] may not directly modulate the physiological processes underlying HRV in the same immediate and measurable way as the insula’s connections with other brain structures, such as the hypothalamus or amygdala.

Moreover, the statistical results indicating no significant correlation between insula -DMN connectivity and HRV, despite the robust insula–DMN functional connectivity, point to the need for exploring additional variables that may mediate this relationship. Several factors that can influence HRV more immediately than brain connectivity should be considered. HRV is known to respond rapidly to stressors, respiration patterns, hormonal fluctuations (e.g., menstrual cycle), and sleep quality [37,38,39,40]. These variables, which modulate HRV almost instantaneously, highlight the complexity of using HRV as a direct marker of brain connectivity, especially when such factors may not correspond directly with the neural activity captured during fMRI sessions. For example, controlled breathing can enhance vagal activity, thereby increasing HRV, without necessarily engaging the brain networks observed through fMRI. It is also worth noting that while insula–DMN connectivity reflects temporal fluctuations observed over the resting state period of acquisition, the HRV analysis relies on a single metric despite having been acquired also at resting state.

Our results highlight the critical role of considering how different brain networks, such as the salience network and the central autonomic network, may integrate with the insula beyond its connection with the DMN. By disentangling these complex interactions, future research may uncover how various brain regions coordinate to support autonomic control and emotional regulation processes.

### Methodological Pitfalls and Future Directions for the Study

Several methodological considerations are crucial for understanding the outcomes. The current study used resting-state fMRI and HRV measurements, focusing on baseline connectivity and autonomic regulation. While resting-state data provide valuable insight into intrinsic functional connectivity, they may lack the specificity required to capture dynamic changes in connectivity that could occur in response to emotional or cognitive tasks. Task-based paradigms, particularly those that elicit emotional or autonomic responses, could reveal whether insula–DMN connectivity is more closely tied to HRV under active, rather than resting, conditions. By incorporating task-based fMRI alongside HRV measurements recorded in real time, future research could examine the situational interplay between brain connectivity and autonomic function more comprehensively. This approach would help elucidate whether the neural networks supporting emotional and autonomic processes respond similarly during rest and in situations that demand emotional regulation, potentially offering a more nuanced understanding of the insula–DMN relationship with HRV.

Conducting longitudinal studies, where participants are measured multiple times over an extended period, can also help identify patterns and reduce variability due to daily fluctuations [41]. Longitudinal designs can provide insights into how brain connectivity and HRV change over time, improving the reliability of findings.

Finally, one significant methodological pitfall of the study is that the fMRI acquisition and HRV measurement were conducted on separate days. Given that HRV can fluctuate rapidly in response to acute stressors, sleep quality, physical activity, and other immediate physiological factors [38,42], measuring HRV on a different day than the fMRI acquisition may not accurately reflect the autonomic state at the time of the brain imaging. In contrast, brain connectivity tends to show less variation over short periods [23]. Consequently, the lack of simultaneity between HRV and fMRI recordings could have obscured potential correlations, as the rapid HRV changes may not have aligned with the neural connectivity patterns captured on a different day. Future studies should aim to collect HRV and fMRI data at resting state concurrently to ensure that the physiological and neural data are temporally synchronized, providing a more accurate picture of their relationship and how they co-vary under similar conditions.

## 5. Conclusions

The hypothesis that stronger insula–DMN connectivity would correlate with higher HRV was not supported by the findings, underscoring the need for further exploration of additional mediating factors and methodological refinements. Although a significant positive correlation was found between insula–DMN connectivity, this relationship did not extend to HRV, suggesting a complex interplay between brain connectivity and autonomic regulation that may require a more intricate model to fully understand. This outcome indicates that, while robust functional connectivity between the insula and DMN exists, it may not directly impact autonomic control as measured by HRV, challenging initial assumptions.

To address the methodological pitfalls identified—such as the temporal mismatch between fMRI and HRV measurements and the need for concurrent data collection—future studies should prioritize simultaneous acquisition of physiological and neural data. Employing task-based paradigms, longitudinal designs, and multimodal approaches will also be essential to capture the temporal dynamics of these systems accurately.

This work sought to elucidate the neural mechanisms underlying brain-autonomic interactions, providing insights into the interplay between emotional regulation, interoception, and autonomic stability. The better elucidation of the connection between brain connectivity and autonomic function could have broader applications, including developing targeted interventions for enhancing emotional regulation. Ultimately, these findings emphasize the necessity for more nuanced and precisely controlled studies to unravel the complex relationship between brain connectivity and autonomic function, thereby contributing valuable insights into emotional regulation mechanisms.

A deeper knowledge of the brain circuits involved in the modulation of cardiac activity can inform therapeutic strategies designed for clinical populations in which these processes are altered. An example is the development of biofeedback interventions targeted to the insula–DMN connectivity through the use of real-time fMRI neurofeedback training or autonomic biofeedback focused on cardiac activity can thus constitute promising applications for clinical use.

## Figures and Tables

**Figure 1 brainsci-15-00037-f001:**
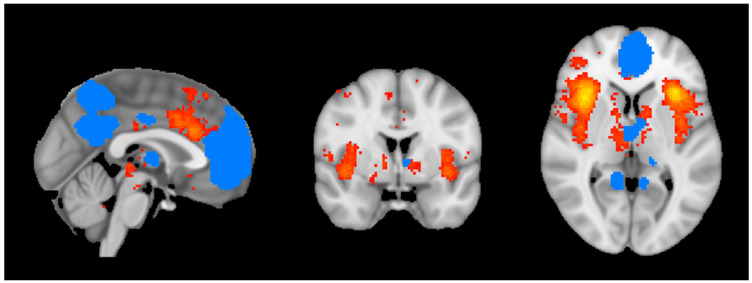
Illustration of selected ROIs for resting state functional analysis, including the insula (in red) and DMN (in blue) for the template subject. ROIs are shown in axial (**right** image), coronal (**center** image), and sagittal (**left** image) planes in neurological convention.

**Figure 2 brainsci-15-00037-f002:**
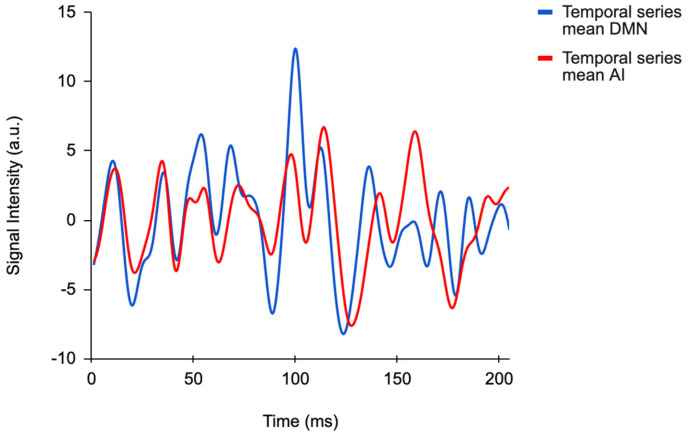
Mean temporal series of functional connectivity of the insula (red line) and the default mode network (DMN).

**Figure 3 brainsci-15-00037-f003:**
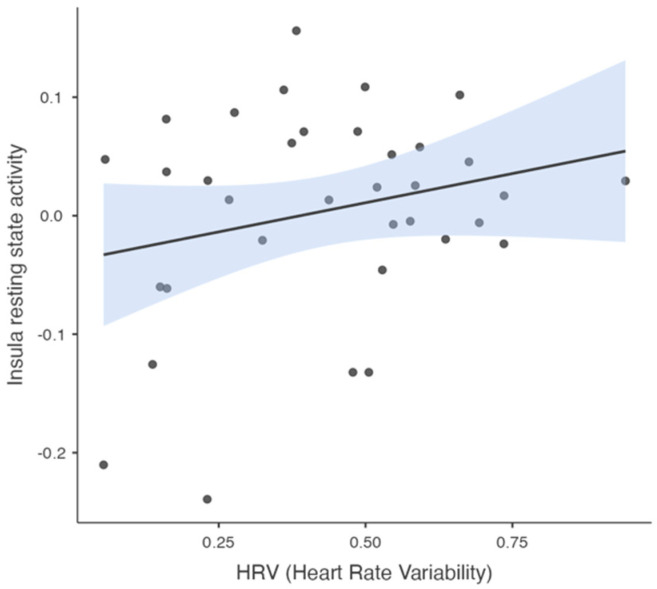
Scatter plot showing the relationship between HRV and insula resting-state activity. The solid line represents linear regression line, with shaded regions indicating the standard error for the predicted values. Each point represents an individual data observation.

**Figure 4 brainsci-15-00037-f004:**
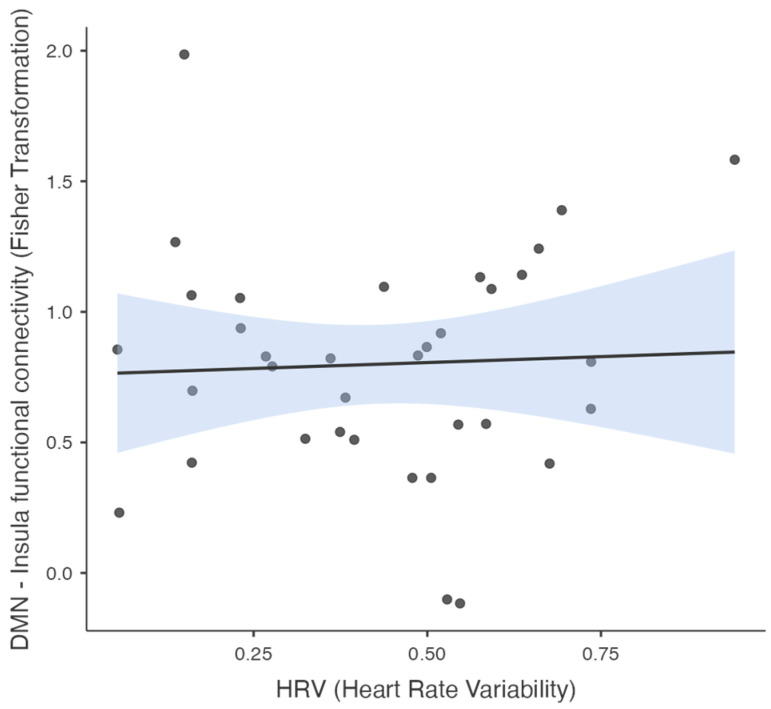
Scatter plot showing the relationship between HRV and insula–DMN functional connectivity. The solid line represents linear regression line, with shaded regions indicating the standard error for the predicted values. Each point represents an individual data observation.

**Table 1 brainsci-15-00037-t001:** Correlation of HRV with insula volumes adjusted for intracranial volumes.

Variable	HRV-HF		
	Pearson Correlation (r)	Sig. (2-Tailed)	N
Right Insula	0.017	0.916	43
Left Insula	0.365	0.016	43
Total Insula	0.202	0.192	43

Note. HRV = heart rate variability. Values are adjusted for intracranial volumes.

## Data Availability

The conditions of our ethical approval do not include the public archiving of anonymized participant data, but data will be made available on request. The data, and code used for their analysis, are available upon request to the corresponding author, with the sole condition that compliance with ethical procedures governing the reuse of the data must be maintained.

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
