# Peer review of "Examining Insula–Default Mode Network Functional Connectivity and Its Relationship with Heart Rate Variability"

_brainsci, 2025, doi:10.3390/brainsci15010037_

Round 1
Reviewer 1 Report
Comments and Suggestions for Authors
Dear Authors,
your study addresses an interesting topic, but certain issues require clarification and refinement to enhance the quality and impact of the manuscript. Overall, the study presents valuable data, but addressing the following points will significantly improve the manuscript’s clarity, coherence, and contribution to the field. Below, I have outlined specific points for improvement:
-
The rationale for why the grey matter volume of the insula should correlate with HRV is not well introduced or supported by the literature. A stronger theoretical explanation is needed to justify Hypothesis 1.
-
In Section 2.7, the sentence “Pearson’s correlation coefficient measures the strength and direction of the linear relationship between two continuous variables” is factually incomplete and misleading, as correlation does not imply causation. Please revise for accuracy.
-
The repeated explanations of insula and DMN functions detract from the manuscript’s focus. Streamline these descriptions, retaining only the details directly relevant to the study’s objectives.
-
Explicitly state how this work advances previous research, such as whether this is the first investigation of Insula-DMN connectivity and HRV or if it addresses specific theoretical or methodological gaps.
-
Elaborate on the clinical or scientific relevance of the findings. Why does this result matter, and what are its broader implications?
-
The discussion of functional connectivity between the insula and DMN is partially redundant and lacks depth. Please refine and expand this section.
-
Discuss the practical implications of your findings to strengthen the manuscript's impact for both research and clinical applications.
Author Response
-
The rationale for why the grey matter volume of the insula should correlate with HRV is not well introduced or supported by the literature. A stronger theoretical explanation is needed to justify Hypothesis 1.
Response: We thank the reviewer for this comment. The introduction elaborates on the relationship between the insula as a major hub of interoception and the HRV as a standard measure of cardiac flexibility in response to interoceptive stimuli. This close involvement argues for the possible association at the volumetric or structural level mentioned in Hypothesis 1. This has been previously reported in other studies (e.g., Matusik et al., 2023). In the revised version of the manuscript, we added one example of these findings and added a few sentences that make the basis for this hypothesis clearer. The changes are highlighted in the revised manuscript (page 2, 89-90 and page 3, line 223-226).
Ref: Matusik, P.S.; Zhong, C.; Matusik, P.T.; Alomar, O.; Stein, P.K. Neuroimaging Studies of the Neural Correlates of Heart Rate Variability: A Systematic Review. J. Clin. Med. 2023, 12, 1016. https://doi.org/ 10.3390/jcm12031016
-
In Section 2.7, the sentence “Pearson’s correlation coefficient measures the strength and direction of the linear relationship between two continuous variables”is factually incomplete and misleading, as correlation does not imply causation. Please revise for accuracy.
Response: Pearson’s correlation coefficient is a statistical measure that quantifies the strength and direction of a linear relationship between two continuous variables. Importantly, as the reviewer rightly stresses Pearson’s correlation captures only linear associations and does not imply causation, as it cannot account for confounding variables, reverse causality, or other underlying factors influencing the relationship. In order to avoid any potential misleading interpretation of the sentence cited by the reviewer we changed it in the revised version of the manuscript deleting the word linear (page 3, line 365).
-
The repeated explanations of insula and DMN functions detract from the manuscript’s focus. Streamline these descriptions, retaining only the details directly relevant to the study’s objectives
Response: We agree that the explanations of insula and DMN functions were a bit repeated and too detailed, so following this suggestion we changed the text so that the introduction is now more focused on the study´s goals. The changes made to the introduction are all highlighted in the manuscript.
-
Explicitly state how this work advances previous research, such as whether this is the first investigation of Insula-DMN connectivity and HRV or if it addresses specific theoretical or methodological gaps
Response: We agree with the reviewer that the final part of the introduction would benefit from a clearer point on how the study advances prior research. Indeed, although prior research has demonstrated links between the insula or the DMN and cardiac function, few studies have explored their combined influence on HRV. We added this part of the text to the revised version of the manuscript (page 3, lines 200-220 and lines 237-241).
-
Elaborate on the clinical or scientific relevance of the findings. Why does this result matter and what are its broader implications?
Response: This point is partially related to the issues raised in the previous point (point 4 related to the scientific relevance of the findings), as well as the next point (point 7 related to the clinical implications/relevance of the findings), so please refer to our answer to those 2 specific questions. We believe that with the changes made in the revised version of the manuscript both the scientific novelty of the study in terms of the theoretical gap it addresses, as well as the possible clinical implications are now clearer for the reader.
-
The discussion of functional connectivity between the insula and DMN is partially redundant and lacks depth. Please refine and expand this section.
Response: We made an effort to eliminate the redundant parts of the discussion in the revised manuscript and deleted some sentences from the text.
-
Discuss the practical implications of your findings to strengthen the manuscript's impact on both research and clinical applications.
Response: We believe we had already mentioned the possible implications of this research for clinical applications, but in the revised manuscript we specified some examples of those possible clinical tools (page 11: lines 521-523 and line 529-534)
Reviewer 2 Report
Comments and Suggestions for Authors
Thank you for the opportunity to review this interesting manuscript on the connection between heart rate variability (HRV) measures and brain activity in the insula and default mode network (DMN). This study is extremely relevant to neuroscience researchers: In fact, a collaborator asked me in the past month specifically about how HRV metrics correlate with brain states! The more is known about the connection between these two important ways physiological indicators of mental health, the more useful both types of measurement will be to researchers across various fields of inquiry. I thought this study was well-explained and well-conducted, with a small but sufficient sample size to draw conclusions. The authors laid out logical reasons for studying the insula and DMN areas in connection with HRV, based on past literature. The acquisition and analysis of data made sense to me, and results were presented very clearly -- I particularly liked the graph that illustrates co-activation of brain areas in Figure 2. The association between left insula volume and HRV suggests that there is something of interest in this relationship, even though the authors' other hypotheses about insula-DMN connectivity or insula activation were not confirmed. I had the same suspicion about reasons for the lack of association that the authors noted in their Limitations section: HRV can be highly variable, and it seems particularly important to measure HRV and fMRI data at the same time if possible, and task-based measures might show a stronger relationship. Still, the current results provide helpful initial data and suggest important directions for future research.
I have two questions that might be interesting for the authors to consider, although neither is really a suggestion for improvement:
1. Given the strong correlation between the insula and the DMN (around r = .50), I wondered at what point the insula should be considered to be simply part of the DMN, rather than a separate subsystem of the brain? I understand that the insula also may have other functions not connected to the DMN, but some other brain areas like the prefrontal cortex seem to have roles in multiple brain subsystems or networks. As brain research has moved away from thinking about localization of certain functions in certain areas, and toward whole-brain networks of co-activation, I wondered where the cut-off might be for considering an area to be part of a network vs. distinct.
2. HRV is commonly regarded as a measure of stress and adaptation, and that seems appropriate in the context of this study -- e.g., HRV is lower when people take longer to calm down from a stressful event, which seems related to the insula's emotion-regulation functions. But HRV is also related to physical fitness (e.g., baseline heart rate) and activity (e.g., exercise). There was some pre-processing of HRV data before the primary analyses, but I wasn't sure whether this pre-processing controlled for baseline HR or other variables that might have an effect on HRV. If not, this might be a way to weed out some of the error variance in the HRV data, and potentially reveal stronger associations between HRV and brain activity measures.
Thanks again for the chance to review this paper. I thought it was well-written and interesting, and hope to see it appear in print soon!
Author Response
-
Given the strong correlation between the insula and the DMN (around r = .50), I wondered at what point the insula should be considered to be simply part ofthe DMN, rather than a separate subsystem of the brain? I understand that the insula also may have other functions not connected to the DMN, but some other brain areas like the prefrontal cortex seem to have roles in multiple brain subsystems or networks. As brain research has moved away from thinking about localization of certain functions in certain areas, and toward whole-brain networks of co-activation, I wondered where the cut-off might be for considering an area to be part of a network vs. distinct.
Response: Indeed, as the reviewer points out brain research has moved from approaches centered on the localization of certain functions in certain areas, toward whole-brain networks of co-activation. This approach was in fact the basis of our hypothesis that the insula activation would be functionally connected with the DMN. In fact, the observed strong correlation between the insula and the DMN aligns with existing literature that highlights the insula's role as a hub of integration between various brain networks. However, we must be aware that the insula is a major hub of another important resting state network which is the salience network. As previous work including our own (da Costa et al., 2022) demonstrated the core regions of the salience network (SN) (like insula) in charge of affective components of empathy and the main nodes of DMN involved in cognitive empathy or mentalizing exhibit an important interplay both during rest and during the execution of specific social-cognitive tasks of Self and Other processing.
As there is no universally agreed-upon "cut-off" for defining network membership, methodological decisions, such as thresholds for functional connectivity strength or network-specific parcellations, should also play a role. For example, Yeo et al.'s 7- and 17-network parcellation (2011) classify the insula variably, with portions associating more strongly with networks other than the DMN. This variability underscores the importance of context and methodology in defining network boundaries.
Ref: Ribeiro da Costa C, Soares JM, Oliveira-Silva P, Sampaio A and Coutinho JF (2022) Interplay Between the Salience and the Default Mode Network in a Social-Cognitive Task Toward a Close Other. Front. Psychiatry 12:718400. doi: 10.3389/fpsyt.2021.718400
-
HRV is commonly regarded as a measure of stress and adaptation, and that seems appropriate in the context of this study -- e.g., HRV is lower when people take longer to calm down from a stressful event, which seems related to the insula's emotion-regulation functions. But HRV is also related to physical fitness (e.g., baseline heart rate) and activity (e.g., exercise). There was some pre-processing of HRV data before the primary analyses, but I wasn't sure whether this pre-processing controlled for baseline HR or other variables that might have an effect on HRV. If not, this might be a way to weed out some of the error variance in the HRV data, and potentially reveal stronger associations between HRV and brain activity measures.